# Age Moderates the Effect of Injury Severity on Functional Trajectories in Traumatic Brain Injury: A Study Using the NIDILRR Traumatic Brain Injury Model Systems National Dataset

**DOI:** 10.3390/jcm11092477

**Published:** 2022-04-28

**Authors:** Laraine Winter, Janell L. Mensinger, Helene J. Moriarty, Keith M. Robinson, Michelle McKay, Benjamin E. Leiby

**Affiliations:** 1Nursing Service, Corporal Michael J. Crescenz Veterans Affairs Medical Center, Philadelphia, PA 19104, USA; helene.moriarty@villanova.edu; 2M. Louise Fitzpatrick College of Nursing, Villanova University, Villanova, PA 19085, USA; michelle.mckay@villanova.edu; 3Department of Clinical and School Psychology, College of Psychology, Nova Southeastern University, Fort Lauderdale, FL 33314, USA; jmensing@nova.edu or; 4Department of Physical Medicine and Rehabilitation, Perelman School of Medicine, University of Pennsylvania, Philadelphia, PA 19104, USA; keith.robinson@va.gov; 5Division of Biostatistics, Department of Pharmacology and Experimental Therapeutics, Sidney Kimmel Medical College, Thomas Jefferson University, Philadelphia, PA 19107, USA; benjamin.leiby@jefferson.edu

**Keywords:** brain injury, traumatic, age, functional impairments, recovery trajectory, injury severity

## Abstract

Age is a risk factor for a host of poor outcomes following traumatic brain injury (TBI), with some evidence suggesting that age is also a source of excess disability. We tested the extent to which age moderates the effect of injury severity on functional trajectories over 15 years post injury. Data from 11,442 participants from the 2020 National Institute of Disability and Independent Living Rehabiitation Research (NIDILRR) Traumatic Brain Injury Model Systems (TBIMS) National Dataset were analyzed using linear mixed effects models. Injury severity was operationally defined using a composite of Glasgow Coma Scale scores, structural imaging findings, and the number of days with post-trauma amnesia. Functioning was measured using the Glasgow Outcomes Scale-Extended. Age at injury was the hypothesized moderator. Race, ethnicity, sex, education, and marital status served as covariates. The results showed a significant confounder-adjusted effect of injury severity and age of injury on the linear slope in functioning. The age effect was strongest for those with mild TBI. Thus, the effects of injury severity on functional trajectory were found to be moderated by age. To optimize outcomes, TBI rehabilitation should be developed specifically for older patients. Age should also be a major focus in TBI research.

## 1. Introduction

Age is a risk factor for a host of poor outcomes following traumatic brain injury (TBI), a major public health problem in the U.S. [1] and globally [2]. Older patients with TBI have a higher mortality [3,4], worse functional outcomes [5,6,7], weaker community reintegration [8], a greater likelihood of re-injury [4], and more emergency department visits compared to younger ones [9].

In addition to its direct effects on important outcomes, some evidence suggests a role for age in excess disability [10,11]. This refers to the phenomenon that some patients’ recovery is worse than would be expected given their relatively mild degree of pathology, whereas others with more severe pathology emerge with better functional outcomes than anticipated. Thus, TBI patients’ functional outcomes may seem disproportionate to their objective level of pathology. Excess disability is commonly observed and has been studied in diverse clinical populations [12,13]. Understanding the factors that account for it has important clinical implications: A factor that accounts for excess disability should be a focus of research and, if possible, intervention. Thus, if the effect of injury severity on functioning depends on the individual’s age at the time of injury, such that an older age is associated with more negative trajectories, efforts to more effectively tailor rehabilitation to the needs of older TBI patients with milder injuries could help to improve rehabilitation outcomes.

The existing research provides evidence of considerable excess disability in TBI and implicates age as a major factor in it. Several studies have reported worse outcomes for older patients, despite their having milder TBIs. For example, Susman et al. [3] found a higher mortality in patients 65 or older with milder injuries compared to younger ones. Marquez de la Plata et al.’s [5] study of age and 5-year functional recovery revealed older patients to have less severe TBI upon admission but worse functional decline subsequently. Livingston and associates [14] reported a worse functional status at discharge and less improvement at one year in TBI patients 60 or older compared with younger ones. In addition, the poorer functional outcomes began to appear even in patients between 45 and 59 years.

In these studies, poorer outcomes occurred despite patients having a milder TBI upon admission. The present study extends that research by explicitly testing whether age moderates the effects of injury severity on recovery trajectories—that is, whether age interacts with injury severity to affect the trajectory of functioning.

The focus on trajectories of functioning rather than outcomes at single points in time further distinguishes the present study from previous research. Functional trajectories are important TBI-related outcomes [15], especially because TBI becomes chronic for many individuals [16]. Indeed, TBI has been called a chronic and even a dynamic condition [8,17,18]. Functional trajectories address the arc of this recovery experience. For this reason, trajectories have a particular clinical relevance in research on aging with TBI. Using the National Institute for Disability, Independent Living, and Rehabilitation Research (NIDILRR) Model Systems National Dataset [19], the present study calculated the trajectories of functional recovery [3,5,8] and examined the role of age at time of injury as a possible moderator of injury severity on these trajectories.

Investigating the role of age in TBI rehabilitation outcomes is especially important in light of the aging of the population, the growing prevalence of TBI in the older age group [20], and their relatively poor outcomes [21]. Dams-O’Connor et al. cited a 20–25 percent increase in U.S. trauma center admissions for TBI among those ≥75 years, relative to the general population, between 2007 and 2010 [22]. These trends have led to a greater recognition of TBI’s importance in the older population [23]. Nevertheless, the effects of age on TBI recovery are still understudied and clinical guidelines underdeveloped [24].

The present study tested the possible moderating effect of age at injury on recovery trajectories up to 15 years post injury. This is a meaningful length of time to capture the change in functioning in chronic TBI. In addition to examining the confluence of injury severity and age of injury on the change in functioning over time, we also tested the prediction that age and severity of injury would be unique sources of change in functioning over time in persons with TBI.

## 2. Method

### 2.1. Participants

The present study utilized data from the NIDILRR Traumatic Brain Injury Model Systems (TBIMS) National Dataset. Than 17,000 were new cases collected in 2020 data base that was sufficiently severe to require hospitalization. All had received comprehensive inpatient rehabilitation services at one of the NIDILRR-funded centers in the United States. Data were collected in person at the time of injury, with follow-up interviews by telephone at one, two, and five years post injury and at five years intervals after that. To be eligible, participants must have experienced an injury-related event (e.g., external mechanical force) and met at least one criterion for moderate or severe TBI (post traumatic amnesia lasting >24 h, abnormal neuroimaging abnormalities, or Glasgow Coma Scale score < 13); been 16 years of age or older at injury; presented to a TBIMS acute care hospital within 72 h of the injury; received both acute hospital care and comprehensive rehabilitation within the TBIMS; and been able to provide consent or have a family member or legally authorized representative who could provide consent. The criteria for inclusion are described more fully elsewhere [25].

The sample size for the present analysis was 11,442. This secondary analysis was approved by the Villanova University Institutional Review Board.

### 2.2. Measures

*Injury Severity.* Classification of TBI severity (mild, moderate, severe) generally takes into account five criteria: Glasgow Coma Scale (GCS) scores [26], structural imaging findings, length of time with loss of consciousness, alternation of consciousness, and post-trauma amnesia [27]. The TBIMS dataset includes data on GCS score, structural imaging findings, and amnesia for some participants but no data for loss of consciousness or alteration of consciousness. Therefore, we used the available criteria to develop an algorithm to estimate injury severity, as follows: Patients who scored 13–15 on the GCS or had post-trauma amnesia (PTA) for less than 1 day and had no structural imaging findings were classified as mild; patients at any level of GCS who did have a positive imaging finding and had PTA for 2–6 days were classed as moderate; and those who had positive imaging findings and PTA lasting longer than 7 days or a GCS score less than 9 were classified as severe.

*Functional Impairment.* The Glasgow Outcomes Scale-Extended (GOS-E) is a 20-item disability scale focusing on the major areas of functioning affected by TBI (e.g., traveling independently; social and leisure activities; disruption in social relationships). It is a self-report measure administered at each follow-up interview. Wilson et al. [28] developed a structured interview to improve on the original GOS, extending the rating categories to eight levels of functional limitations, on which 8 = “no functional limitations.” Deceased is coded 0. Therefore, higher scores indicate better functioning.

*Sociodemographic characteristics.* Sex, race, years of education, and marital status were examined as possible covariates. Race was coded White vs. non-White, and marital status as married vs. not married (single, divorced, separated, widowed).

### 2.3. Procedure

Initial interviews were held in the in-patient facilities where patients were treated for their TBI. Follow-up interviews were conducted by telephone.

*Data Analysis.* Data were analyzed using SAS 9.4 (SAS Institute, Cary, NC, USA). After performing descriptive statistics, linear mixed effects models were used to address our primary aims. First, we fit a series of unconditional growth models using the methods outlined by Singer and Willet [29] to establish the best curvature of the slope for functioning over the 15-year post-acute time period. The final model included the following fixed effects: a linear time effect (represented by years post-injury); a quadratic time effect (represented by years post-injury squared), which together represent the nonlinear form of the trajectory of change in functioning; the age of injury (representing the effect of age on functioning at the beginning of the post-acute phase); the injury severity (representing the effect of injury severity on functioning at the beginning of the post-acute phase); an age by injury severity interaction (representing the differential effects of severity on functioning by age at the beginning of the post-acute phase); an age by linear time and an age by quadratic time effect (representing the effect of age on the change in functioning over time); and an age by injury severity by linear time interaction effect and an age by injury severity by quadratic time interaction effect (representing the confluence of age and injury severity on the change in functioning over time). We also adjusted for the effects of race, marital status, education, and sex on the initial status in functioning. Random effects for the intercept and the linear slope were included in the model to allow for subject-specific differences in the initial status for functioning (represented by intercept variance) and the trajectories of change (represented by the slope variances).

Mixed effects models are an especially advantageous analytic framework for questions about the predictors of change trajectories because random effects allow one to model person-specific differences in slopes as the outcome of interest. In addition, mixed models use maximum likelihood estimation, which allows for the maintenance of all observations with complete data on the predictor variables in the analysis as long as they have at least one data point to contribute to the outcome variable. This makes them preferable to repeated measures ANOVA-methods, which use listwise deletion when a case is missing any observations on the outcome.

## 3. Results

Table 1 presents the sociodemographic characteristics of the participants in the study sample at the time of injury.

In reference to the primary aim, the findings revealed a significant confounder-adjusted effect of age of injury on the linear slope in functioning over the years beyond the post-acute treatment phase. The effect of age was strongest for those with mild TBI, γ = −0.0037, SE = 0.0004 *t* = −8.51, *p* < 0.001. For those with moderate TBI (GCS scores between 9 and 12) and severe TBI (GCS < 8), the effect of age was less pronounced, γ = −0.0028, SE = 0.0005 *t* = −5.28 *p* < 0.001; and γ = −0.0024, SE = 0.0003 *t* = −6.95 *p* < 0.001, respectively, showing that the impact of injury on age is less evident for those with more severe injuries. The estimated marginal means of functioning over time for age of injury, ranging from 20 years to 80 years, in 10-year increments are shown in Figure 1, panels A, B, and C for mild, moderate, and severe injuries, respectively.

The quadratic slope also differed by age of injury in all severity groups, meaning that the curvature of the trajectory of functioning differed by age. As an inspection of Figure 1 indicates, older age led to a steeper decline in all severity groups. Inspection of the figure panels indicates that age effects are manifested across the age span, with the trajectory of recovery sloping downward by age 50 at all severity levels.

The effect was again strongest in the mild group (γ = 0.0002, SE = 0.00004 *t* = 4.52, *p* < 0.001) and smaller in the moderate and severe TBI groups (γ = 0.00011, SE = 0.00005 *t* = 2.27 *p* = 0.024; and γ = 0.00010, SE = 0.00003 *t* = 2.87 *p* = 0.004), respectively. Table 2 presents confounder-adjusted model estimates, presented in the Singer and Willet format [29].

The analysis also revealed significant main effects for injury severity and age, as well as the predicted interaction effect of age injury severity, as noted. When an interaction is detected using this method, however, a main effect cannot be interpreted independently, making it difficult to make global statements about the effect of the interacting variables. Nevertheless, as is clear in the plot, there was little difference between the mild and moderate groups at baseline regardless of age. If anything, the moderate group might have had slightly higher GOS-E scores than the mild group, whereas the severe group was always lower than the other groups at baseline, regardless of age.

## 4. Discussion

The study findings confirm that the effect of injury severity on functional trajectory is moderated by age. Thus, the predictive validity of injury severity for functional trajectories is conditional on the individual’s age at the time of injury, with older people experiencing worse functional trajectories. This indicates that excess disability (i.e., weak associations between injury severity and functional outcomes) is heavily attributable to age.

These findings are consistent with previous research documenting age effects on outcomes at specific points in time. The present study extends those findings to functional trajectories [3,4,5,9] as the outcome of interest and confirms the moderation-by-age hypothesis. In addition, the present data confirm the finding that age effects are already manifest in middle age [14,30]. Thus, the adverse effects of age are not limited to the oldest age groups.

How can we account for the moderating effect of age on functional trajectories? The steeper downward functional trajectories are likely to reflect normal aging processes or comorbidities of aging. In the absence of data from a non-TBI population, there is no way to assess this. In addition, data on comorbidities that may impair functioning were not available and therefore could not be examined as mediators or moderators. For example, the presence of TBI is associated with the development of neurodegenerative disease later in life [19], which may affect future functional decline. The GOS-E scale measures disability in independence, social activities, and social relationships. Research on specific areas of functioning decline measured within the GOS-E in older adults with mild TBI (mTBI) may provide further insight into where rehabilitation services should be targeted to improve functioning or prevent decline as they age.

A potential factor in age-associated excess disability may be frailty. This has various definitions [31], with common components being weakness, slowness, exhaustion, low activity, and weight loss [32]. Frailty affects fifteen percent of community-dwelling persons over 65 years of age, with another forty-five percent considered pre-frail [33]. Frailty influences not only a person’s response to injury but also the response to treatment and resulting outcomes. Could older adults with mTBI (in addition to those with moderate and severe) have frailty contributing to falls that lead to more functional decline over time? In Abdulle and associates’ [34] study of 161 older adults with mild TBI, fewer than a quarter fully recovered from their injuries. A majority of the frail older adults were left with a significantly worse disability compared to the non-frail.

Despite its role as a risk factor for poor rehabilitation outcomes, frailty is not always assessed clinically. Frailty is thought to be reversible, especially among those considered pre-frail [35]. In adults with TBI, the assessment of frailty at time of injury and continuously throughout the post-injury phase may be useful in predicting functional decline. Future research should investigate whether frailty screening and frailty-focused intervention can mitigate functional decline or excess disability. If so, targeted interventions may improve frailty and prevent its progression, decreasing excess disability in the TBI population. Indeed, the International Initiative for Traumatic Brain Injury Research Consortium of leading health-care professionals has argued for further understanding of the relationship between frailty and TBI [36].

A caveat when interpreting the relatively steep downward trajectories for older individuals is that the GOS-E includes several questions about employment. Retirement, more common among older individuals than younger ones, may produce spuriously low GOS-E scores for older individuals, suggesting that their functioning is lower than it actually is. A second caveat is that all individuals with mTBI in the TBIMS sample had received inpatient rehabilitation services, suggesting that they may have more problems than typical mTBI patients. Their GOS-E scores may be lower than expected in persons with mTBI.

Although older individuals overall showed worse functioning over time in all levels of TBI severity, especially steep declines were observed among those with mTBI. This was an unexpected finding. Several lines of speculation are possible. First, geriatric patients with mTBI may have received less inpatient rehabilitation than older ones with moderate and severe TBI [37]. Since inpatient rehabilitation is important for maximizing recovery, provider bias against rehabilitation for older individuals should be minimized [38]. This argues for specialized neurologic rehabilitation services for this subgroup to prevent the excessive disability that is documented in the present study.

A further consideration is that some comorbidities, especially PTSD and depression, occur more commonly in mild TBI than in moderate or severe. Clinical depression is more common after a mild TBI than after moderate or severe in patients of all ages [38,39]. PTSD is also more common in mTBI than in moderate or severe [40,41]. Such mental health sequelae of mTBI could contribute to functional decline over time, and their predominance in mTBI may help account for the worse functional trajectories in that group of older individuals. Another speculation is that many TBIs in older adults may be misdiagnosed as milder than they really are, perhaps owing to missed structural imaging findings (i.e., false negatives). Such missed findings might occur in older individuals because of age-associated biological changes such as a decrease in brain volume that may mask hemorrhaging in the brain. Thus, some patients with more severe injuries may be misclassified as mTBI [42].

*Clinical Implications.* Standard rehabilitation may be less beneficial for older patients than younger ones due to changes attributable to normal aging and comorbidities common in older age. Geriatric rehabilitation for TBI should differ from standard rehabilitation in specific ways [42,43,44]. Allowing a longer period of time may accommodate a slower trajectory of improvement. Rehabilitation may require more repetition and sustained rehearsal, especially for patients with cognitive impairment. Maintenance rehabilitation may be needed, geared toward sustaining optimal functioning over time. Activities may be directed by family caregivers, as well as self-directed.

Falls are also a major issue [9] both because they are a major cause of TBI in older adults and because of the likelihood of *recurring* falls. Medications that cause hypotension or dizziness or are dopaminergic may increase the risk of falling and should be avoided. Anticoagulant use increases the risk of serious bleeding in falls in older adults with TBI [43]. Poor balance also increases the risk of falls and should be assessed and remediated. Patients’ fear of falling may have serious implications for activity engagement, risk of deconditioning, and recurring falls. A study by Bandeen-Roche and colleagues [33] revealed that half of frail older adults in the U.S. had experienced a fall in the previous year. Frailty might therefore amplify the excess disability seen in TBI.

Findings also highlight the needs of those with chronic TBI for some form of rehabilitation. By five years post injury, little rehabilitation is typically being delivered to patients [42]. Those who need continuing help are a neglected population. Rehabilitation treatments that continue into the chronic phase should be designed, tested, and implemented [45].

*Research Implications.* The present findings raise the question of why older patients with TBI fare worse than their injury severity would predict. The high prevalence of comorbidities in older adults with TBI may account for it. Such comorbidities are a powerful negative prognostic indicator of recovery, as several studies TBI have shown [46,47]. Kumar and colleagues’ [48] study of 393 TBI patients with moderate and severe TBI evaluated the impact of physical, mental, and total health burden on functional and life satisfaction up to 10 years post injury. Their findings demonstrated the long-term impact of comorbidity on recovery from injury. Perhaps comorbidity burden or particular comorbidities mediate the effect of age on functional recovery. The TBI Model Systems Database does not include data on comorbidities that would allow additional analyses with these variables. Until mediators of age effects on TBI recovery are identified, it will be difficult to know how to intervene clinically. Therefore, a promising direction for future research should be the mediation analyses of comorbidities or other factors that could explain why age affects TBI outcomes.

## 5. Conclusions

The growing number of older adults in the population and their increasing representation in the TBI population highlight the importance of age as a crucial factor in TBI outcomes. These trends underscore the need for further attention to age effects on TBI recovery. This should be a major focus of research and clinical attention in the future.

## Figures and Tables

**Figure 1 jcm-11-02477-f001:**
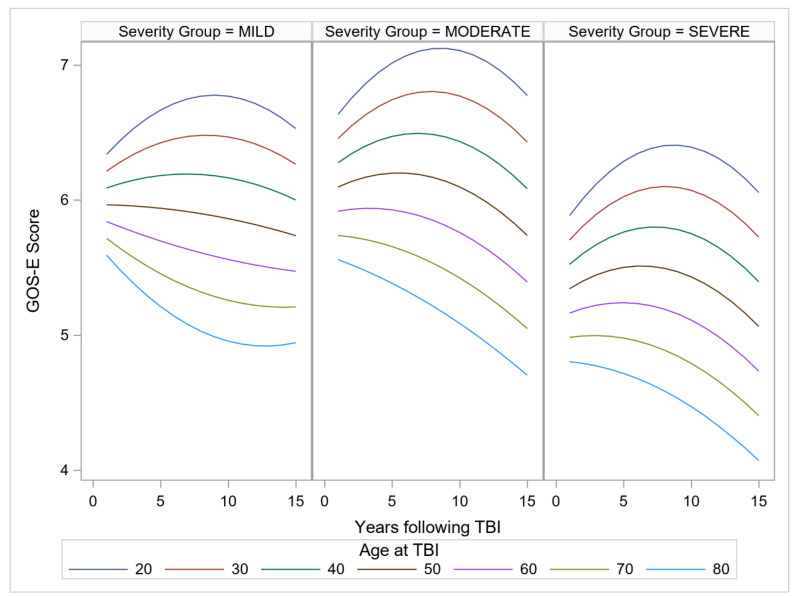
Functioning over time for age of injury in 10-year increments from 20 to 80 years, with the three panels representing mild, moderate, and severe injuries, respectively. TBI: traumatic brain injury; GOS-E: Glasgow Outcomes Scale-Extended.

**Table 1 jcm-11-02477-t001:** Sociodemographic characteristics of sample at year 1 post injury (n =11,442).

	Mean (SD)	Range	Percent (n)
Age	40.6 (18.7)	16–88	
Sex (% male)			73.6 (8420)
Race (% White)			69.7 (7860)
Hispanic ethnicity (% Hispanic)			10.3 (1177)
Education (number of years)	12.5 (2.9)	1–20	
Marital status (% married)			33.9 (3972)

SD: Standard deviation.

**Table 2 jcm-11-02477-t002:** Confounder-adjusted mixed-effects model predicting functioning over time.

	γ (SE)	*p*
Fixed effects		
Model for initial status at 1 year post-acute injury		
Intercept	5.254 (0.076)	<0.001
Age at injury	−0.015 (0.001)	<0.001
Injury severity (mild)	0.691 (0.065)	<0.001
Injury severity (moderate)	0.775 (0.069)	<0.001
Injury severity (severe) [reference]	-	-
Age × injury severity (mild)	0.0056 (0.0025)	0.026
Age × injury severity (moderate)	0.00009 (0.0027)	0.972
Age × injury severity (severe) [reference]	-	-
Covariates		
Racial and/or ethnic minority	−0.407 (0.032)	<0.001
Non-Hispanic white [reference]	-	-
Female	−0.111 (0.033)	<0.001
Male [reference]	-	-
Married	0.120 (0.034)	<0.001
Nonmarried [reference]	-	-
Years of education	0.105 (0.005)	<0.001
Model for linear slope		
Intercept	0.185 (0.013)	<0.001
Age × injury severity (mild)	−0.0037 (0.0004)	<0.001
Age × injury severity (moderate)	−0.0028 (0.0005)	<0.001
Age × injury severity (severe)	−0.0024 (0.0003)	<0.001
Model for quadratic slope		
Intercept	−0.011 (0.0013)	<0.001
Age × injury severity (mild)	0.00020 (0.00004)	<0.001
Age × injury severity (moderate)	0.00011 (0.00005)	0.024
Age × injury severity (severe)	0.00010 (0.00003)	0.0041
Random effects		
Level 1	**σ^2^ (SE)**	
Within-person	0.946 (0.032)	
Level 2	**τ^2^ (SE)**	
Initial status	1.894 (0.037)	
Linear slope	0.0030 (0.00047)	

Notes. Total sample size after adjustments for covariate missingness, N = 11,442. SE: Standard error.

## Data Availability

Data supporting this analysis were from the NIDILRR TBI Model Systems dataset.

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
