# Peer review of "Age Moderates the Effect of Injury Severity on Functional Trajectories in Traumatic Brain Injury: A Study Using the NIDILRR Traumatic Brain Injury Model Systems National Dataset"

_jcm, 2022, doi:10.3390/jcm11092477_

Round 1
Reviewer 1 Report
This topic is interesting, even if paper is hard to read. Some revisions are needed. Look at these points:
- The whole introduction needs to be better organized, as different sentences are often not sequential.
- Lines 39-48: "For example, Susman et al.3 found higher mortality... five years after TBI." These sentences seem part of discussion, please move it down.
- Lines 24-26: "Older patients with TBI have higher mortality, worse functional outcomes, weaker community reintegration, a greater likelihood of re-injury... " As also the impact force and the presence of intracranial hemorrhagic lesions. Please, look at these refs: -- Posttraumatic synchronous double acute epidural hematomas: Two craniotomies, single skin incision. Surg Neurol Int. 2020 Dec 11;11:435. doi: 10.25259/SNI_697_2020. -- Patient-Reported Problem Areas in Chronic Traumatic Brain Injury. J Head Trauma Rehabil. 2021 Nov 2.
- Line 141: "Age of injury was also a predictor of the quadratic slope in functioning in all severity groups" What do authors mean with this sentence? Please revise.
- Table 2 appears to show that age, but also the severity of the injury affects the outcome. Discuss more in the discussion section.
- Lines 179-183: "Yet, frailty is not always assessed clinically despite its important role as a risk factor for poor rehabilitation... thought to be reversible, especially among those... " These data were taken from the literature, what did the authors find about this point? Please report in discussion section.
- Lines 236-237: ""Until mediators of age effects on recovery are identified, it will be difficult to identify clinical implications adequately". Is this a limitation of the paper? Highlight it better.
- Lines 239-241: "Dams-O’Connor et al. cited a 20–25 percent increase in U.S. trauma center admissions for TBI among those > 75 years, relative to the general population, between 2007 and 2010" Why this paper in your conclusion section? Move it the discussion section. Write here what your paper add to the literature.
- Minor: In table 2. Please report p value. As for example p = 0.05 ?
Author Response
This topic is interesting, even if paper is hard to read. Some revisions are needed. Look at these points:
- The whole introduction needs to be better organized, as different sentences are often not sequential.
- Lines 39-48: "For example, Susman et al.3 found higher mortality... five years after TBI." These sentences seem part of discussion, please move it down.
These papers are the core of the literature review, laying the groundwork for the present study. They describe previous research on which our study expands by (a) focusing on recovery trajectories and (b) explicit testing of an interaction between age and injury severity. We therefore consider these references essential to the Introduction. It is not clear to us why the reviewer considers them appropriate for the Discussion section.
- Lines 24-26: "Older patients with TBI have higher mortality, worse functional outcomes, weaker community reintegration, a greater likelihood of re-injury... " As also the impact force and the presence of intracranial hemorrhagic lesions. Please, look at these refs: -- Posttraumatic synchronous double acute epidural hematomas: Two craniotomies, single skin incision. Surg Neurol Int. 2020 Dec 11;11:435. doi: 10.25259/SNI_697_2020. -- Patient-Reported Problem Areas in Chronic Traumatic Brain Injury. J Head Trauma Rehabil. 2021 Nov 2.
We have carefully reviewed both papers. The first was a case study of a 34 year old woman who underwent single skin incision and craniotomies for a fatal traumatic double acute EDH. We are unable to see the connection between this case and our study of age as a moderator of the effect of severity on recovery trajectories. The same is true for the second study, on which Dr. Winter is a co-author. We cannot see the connection between this study’s findings and the current paper and therefore are unsure how and why to cite them.
- Line 141: "Age of injury was also a predictor of the quadratic slope in functioning in all severity groups" What do authors mean with this sentence? Please revise.
We have revised as follows:
“The quadratic slope also differed by age of injury in all severity groups, meaning that the curvature of the trajectory of functioning differed by age. As inspection of the Fig. 1 indicates, older age led to steeper decline in all severity groups.”
- Table 2 appears to show that age, but also the severity of the injury affects the outcome. Discuss more in the discussion section.
In the Results section we have added the following explanation of the main effects and their interpretation:
“The analysis revealed significant main effects for injury severity and age, as well as the predicted interaction effect of age X injury severity, as noted. When an interaction is detected, a main effect cannot be interpreted independently, making it difficult to make global statements about the effect of the interacting variables. Nevertheless, as is clear in the plot, there is not much difference in mild and moderate groups at baseline regardless of age (if anything, the moderate group might have slightly higher GOS-E scores than the mild group), while the severe group is always lower than the other groups at baseline, regardless of age.”
- Lines 179-183: "Yet, frailty is not always assessed clinically despite its important role as a risk factor for poor rehabilitation... thought to be reversible, especially among those... " These data were taken from the literature, what did the authors find about this point? Please report in discussion section.
We cited several published reviews on TBI in the older adult population. The authors of these reviews based this observation on literature that they reviewed. In our Discussion, we proposed that, based on the evidence these authors presented, frailty may help to explain excess disability. We therefore discussed the implications of this for practice and assessment. We have tightened this up to make it these implications clearer.
- Lines 236-237: ""Until mediators of age effects on recovery are identified, it will be difficult to identify clinical implications adequately". Is this a limitation of the paper? Highlight it better.
This is not a limitation of the present paper but an important direction for future research. It was not the aim of the paper to test this and is beyond its scope. Our finding that age moderates the effect of injury severity on recovery trajectories would suggest that an explanation for the effect of age on outcomes should be sought in future research, which would have to be conducted using data that includes comorbidity information.
- Lines 239-241: "Dams-O’Connor et al. cited a 20–25 percent increase in U.S. trauma center admissions for TBI among those > 75 years, relative to the general population, between 2007 and 2010" Why this paper in your conclusion section? Move it the discussion section. Write here what your paper add to the literature.
We agree with the reviewer about the placement of this section. We have moved this sentence to the Introduction. We have also made a clearer statement about what our study adds to the literature.
- Minor: In table 2. Please report p value. As for example p = 0.05 ? We have not found any missing p values in Table 2. Reference values are not reported.
Reviewer 2 Report
The authors have thoroughly revised the manuscript, which is now much more understandable and provides important new data on the interaction between age and TBI. The limitations have been discussed sufficiently.
Author Response
Reviewer 2 did not make specific comments.
Round 2
Reviewer 1 Report
Good.
This manuscript is a resubmission of an earlier submission. The following is a list of the peer review reports and author responses from that submission.
Round 1
Reviewer 1 Report
This is an interesting paper examining the role of age on the long-term recovery trajectory after a TBI. The topic is extremely relevant, especially because of the steeply rising numbers of TBIs in the elderly, and because of the aging population in general. The paper is well written and easy to follow, and the methodology used seems valid and sound. The limitations have been addressed quite well. I still have some mainly minor comments:
- One of the key findings is the interaction of age and long-term outcome in patients with mild TBI (mTBI). While the authors state that all these have received in-patient rehabilitation, their mTBI may have been more severe than is usually regarded as an mTBI. What remains a bit unclear is how the severity has been classified. The authors write: “it (=GCS) was determined at the in-patient rehabilitation facility where the patient was treated”. Does this mean that the level of consciousness used in this classification was not based on the GCS at hospital admission, but at some later point when they were transferred to rehabilitation from acute care? Please clarify. If this has indeed been the case, then the whole severity classification is of questionable value.
- Doesn’t the dataset include information about neurosurgery or concomitant injuries? Those with higher age have higher frequency of intracranial bleedings due to many reasons, and thus more often need e.g. hematoma surgery. If data on surgical measures or other injuries (e.g. ISS-scores) is available, that should be added as a covariate. Please add clarifications for these, and possibly further analyses if available.
- The authors could add a bit more discussion on the reliability of the severity classification in different age groups. There is evidence that the GCS (or PTA) do not behave similarly in elderly people as in those of working age. Because of atrophy and ither structural age-related changes GCS 15 may be a different thing in an octogerian compared to a young person. Fairly often elderly people show intracranial bleeding despite a high GCS level.
- The authors discuss the effect of frailty but do not discuss the effect of an underlying neurodegenerative process, although different types of dementia become increasingly prevalent by age. Fairly often it appears that the expected recovery from an mTBI does not take place because the injury makes visible the dormant dementia process, with a subsequent progressive deterioration.
- What remains unclear from the paper is the time frame of the clinical follow-up. The authors refer to an earlier paper, but main information should be provided here, too. During which period were these persons injured? How long was the follow-up (on the average and range)? Has GOSE been applied in the similar manner all the time? Does it consider disability because of extracranial trauma or only the TBI? How has this been confirmed in telephone interviews? Do the authors consider a telephone interview as a limitation? Could it be one reason why those with more severe injuries have better outcome? – they have more problems with self-awareness.
- Marital status married vs. not married – increasingly people live together without being officially married – are these married or not?
- In the Methods ethnicity has not been described.
- It is quite striking and unexpected – as the authors state – that those with mTBI have actually the worst outcomes, as Figure 1 shows. It appears that from those injured at their 40’s, those with mTBI end up to the lowest GOSE level, and the older the age at injury, the lower will be the GOSE at 30 yrs from the injury. Isn’t this simply because those who are 50 yrs or more at injury, will be 80 or more after 30 yrs. Probably none of those who sustain their trauma at the age of 80 will live 30 yrs. How have deaths been taken into account in the trajectories? If those who have died have been excluded, can it be that among those who have sustained moderate-severe TBIs many have died during the follow-up, and those who have stayed alive have either had not so severe injuries (GCS is a very rough measure and influenced by many things) or have otherwise good general health, or both? Please discuss this finding even more, because without good explanations it does not make sense.
- The authors discuss the GOSE and retirement. If the patient had been at work after the injury, but later retired, how has this been taken into account in the grading? Could this make any bias?
- Was the duration of inpatient rehabilitation included in the database? If yes, could this give more information of the true severity than GCS?
Reviewer 2 Report
The intent of the manuscript is good. I believe the analysis as described also is sound (But is Fig 1produced in SPSS ? looks more like a ggplot2 figure from R - in case citation/references should be added).
What I simply do not believe is Fig 1 - the magnitude of difference between GOSE according to severity is very small at time 0. Moreover, the trajectories for functioning are quite similar for all agegroups apart from 60, 70 and 80 years old at TBI. I have sincere doubts about the number of data points allowing the extrapolation of results to 30 years following TBI in an 80-year old at the time of injury! Moreover, there is only a brief table giving demographics. A larger table 1 would be better and perhaps could lend support to the claims that I find highly troublesome. The repercussions would be that there is hardly any significance of the injury severity on outcome - An assertion that is at odds with most literature.
The manuscript is otherwise well written, but rather verbose and could benefit from shortening of both the introduction and discussion.